# Thalamic regulation of ocular dominance plasticity in adult visual cortex

Yi Qin[1,2], Mehran Ahmadlou[3], Samuel Suhai[1], Paul Neering[1], Leander de Kraker[1], J Alexander Heimel[3], Christiaan N Levelt[1,4]*

[1]Molecular Visual Plasticity Group, Netherlands Institute for Neuroscience, Royal Netherlands Academy of Arts and Sciences, Amsterdam, Netherlands; [2]University of Strasbourg, Strasbourg, France; [3]Circuits, Structure and Function Group, Netherlands Institute for Neuroscience, Royal Netherlands Academy of Arts and Sciences, Amsterdam, Netherlands; [4]Department of Molecular and Cellular Neurobiology, Center for Neurogenomics and Cognitive Research, VU University Amsterdam, Amsterdam, Netherlands

*For correspondence:
c.levelt@nin.knaw.nl

## Abstract

Experience-dependent plasticity in the adult visual system is generally thought of as a cortical process. However, several recent studies have shown that perceptual learning or monocular deprivation can also induce plasticity in the adult dorsolateral geniculate nucleus (dLGN) of the thalamus. How plasticity in the thalamus and cortex interact in the adult visual system is ill-understood. To assess the influence of thalamic plasticity on plasticity in primary visual cortex (V1), we made use of our previous finding that during the critical period ocular dominance (OD) plasticity occurs in dLGN and requires thalamic synaptic inhibition. Using multielectrode recordings we find that this is also true in adult mice, and that in the absence of thalamic inhibition and plasticity, OD plasticity in adult V1 is absent. To study the influence of V1 on thalamic plasticity, we silenced V1 and show that during the critical period, but not in adulthood, the OD shift in dLGN is partially caused by feedback from V1. We conclude that during adulthood the thalamus plays an unexpectedly dominant role in experience-dependent plasticity in V1. Our findings highlight the importance of considering the thalamus as a potential source of plasticity in learning events that are typically thought of as cortical processes.

## eLife assessment

This study demonstrates that plasticity of ocular dominance of binocular neurons in the visual thalamus persists in adulthood. The evidence supporting the authors' conclusion is **convincing**, and the findings are an **important** contribution to a growing body of work identifying plasticity in the adult visual system. This work will interest those in the field of ocular dominance plasticity in the visual system as well as scientists investigating the function of synaptic plasticity in the brain.

## Introduction

Experience-dependent plasticity in the adult visual system is largely thought of as a cortical process (*Gilbert et al., 2009*; *Gilbert and Wiesel, 1992*). However, several recent studies have demonstrated that plasticity also occurs in the adult dorsal lateral geniculate nucleus (dLGN) of the thalamus of human subjects during perceptual learning (*Yu et al., 2016*) and mice upon monocular deprivation (MD) (*Huh et al., 2020*; *Jaepel et al., 2017*). How plasticity in adult dLGN is regulated and whether plasticity in dLGN and V1 influence each other have barely been studied.

In this study, we addressed these questions using ocular dominance (OD) plasticity in mice as a model. OD is the property that neurons preferentially respond to visual stimuli presented to one eye versus the other (*Wiesel and Hubel, 1963a*). Visual experience affects OD and a period of MD results in an OD shift in V1 neurons due to weakened responses to the deprived eye and strengthened responses to the non-deprived eye (*Hensch and Stryker, 1996*; *Wiesel and Hubel, 1963a*). OD plasticity is most prominent during a critical period of development (*Gordon and Stryker, 1996*; *Wiesel and Hubel, 1963a*) but can also be induced in young adult mice (*Heimel et al., 2007*; *Hofer et al., 2006*; *Lehmann and Löwel, 2008*; *Sato and Stryker, 2008*; *Sawtell et al., 2003*). This requires a longer period of deprivation, however, and the shift is smaller and less persistent and is mediated predominantly by strengthening of responses to the non-deprived eye (*Frenkel and Bear, 2004*; *Heimel et al., 2007*; *Hofer et al., 2006*; *Lehmann and Löwel, 2008*; *Sato and Stryker, 2008*).

Previously, we demonstrated that during the critical period extensive OD plasticity can be induced in dLGN and that this requires synaptic inhibition in the thalamus (*Sommeijer et al., 2017*). Multielectrode recordings revealed that OD plasticity in dLGN is strongly reduced in mice in which thalamic synaptic inhibition is inactivated by deleting the gene encoding the GABA receptor alpha1 subunit (*Gabra1*) selectively in the dorsal thalamus (*Gabra1*$^{fl/fl}$ × *Olig3*$^{Cre/+}$ mice, hereafter referred to as '*Gabra1* cKO mice'). Interestingly, OD plasticity induced by long-term MD is also reduced in V1 of these mice due to the absence of ipsilateral eye response strengthening (*Sommeijer et al., 2017*), suggesting that during the critical period thalamic plasticity contributes to plasticity in V1. Here, we investigated how dLGN and V1 influence each other during OD plasticity in adulthood. We find that in adult mice lacking thalamic synaptic inhibition OD plasticity is absent in both dLGN and V1. Silencing V1 of adult wild-type (WT) mice does not affect the OD shift in dLGN, showing that it does not depend on feedback from V1. In contrast, we find that during the critical period the OD shift in dLGN partially depends on activity in V1. Together, our findings show that thalamocortical interactions underlying OD plasticity change with age and suggest that the thalamus may be an important source of plasticity in adult learning events that have generally been considered cortical processes.

## Results

### Visual responses in adult dLGN of WT and *Gabra1* cKO mice

Previous work showed that during the critical period visual responses of dLGN neurons in *Gabra1*$^{fl/fl}$ × *Olig3*$^{Cre/+}$ (*Gabra1* cKO) mice were less sustained due to the lack of thalamic synaptic inhibition, while average response strengths and basic receptive field properties seemed surprisingly unaffected (*Sommeijer et al., 2017*). To assess whether this situation remained similar in adulthood, we measured visual responses in dLGN using a 16-channel silicon probe in anesthetized *Gabra1* cKO mice and *Gabra1*$^{fl/fl}$ × *Olig3*$^{Cre/-}$ (WT) siblings. Recordings were performed in the ipsilateral projection zone of dLGN (*Figure 1A*). Receptive field sizes and positions were determined by presenting white squares (5°) at random positions on a black background (*Figure 1B*). We only included recordings from channels with receptive fields corresponding to the central 30° of the visual field. We observed no differences in receptive field sizes in *Gabra1* cKO and WT mice (*Figure 1C*). As receptive field sizes in dLGN are known to become smaller between eye opening and critical period onset (*Tschetter et al., 2018*), this observation suggests that dLGN develops surprisingly normally in the absence of synaptic inhibition. To investigate this further, we analyzed the numbers, densities, and sizes of inhibitory and cholinergic boutons, which are also known to increase during the same developmental window (*Bickford et al., 2010*; *Sommeijer et al., 2017*; *Sokhadze et al., 2018*). Again, in adult WT and *Gabra1* cKO mice we observed no differences (*Figure 1—figure supplement 1*).

To record visual responses of dLGN neurons to the contra- or ipsilateral eye separately, the other eye was closed and visual stimuli (full screen, full contrast black/white reversals, at 3 s intervals) were presented (*Figure 1B*). We selected single units from non-deprived WT and *Gabra1* cKO mice and assessed their responses to the contra- and ipsilateral eye. Examples of monocular and binocular single units are shown in *Figure 1D*. In WT mice, 51% of recorded neurons were binocular and in *Gabra1* cKO mice 59%. To assess whether the temporal profile of visual responses differed in adult WT and *Gabra1* cKO mice, we compared the area under the curve (AUC) of the peristimulus time histogram (PSTH) during different time bins (*Figure 1E*). This revealed that in *Gabra1* cKO mice visual responses attenuated faster than in WT siblings: peak responses were stronger, but weaker responses were

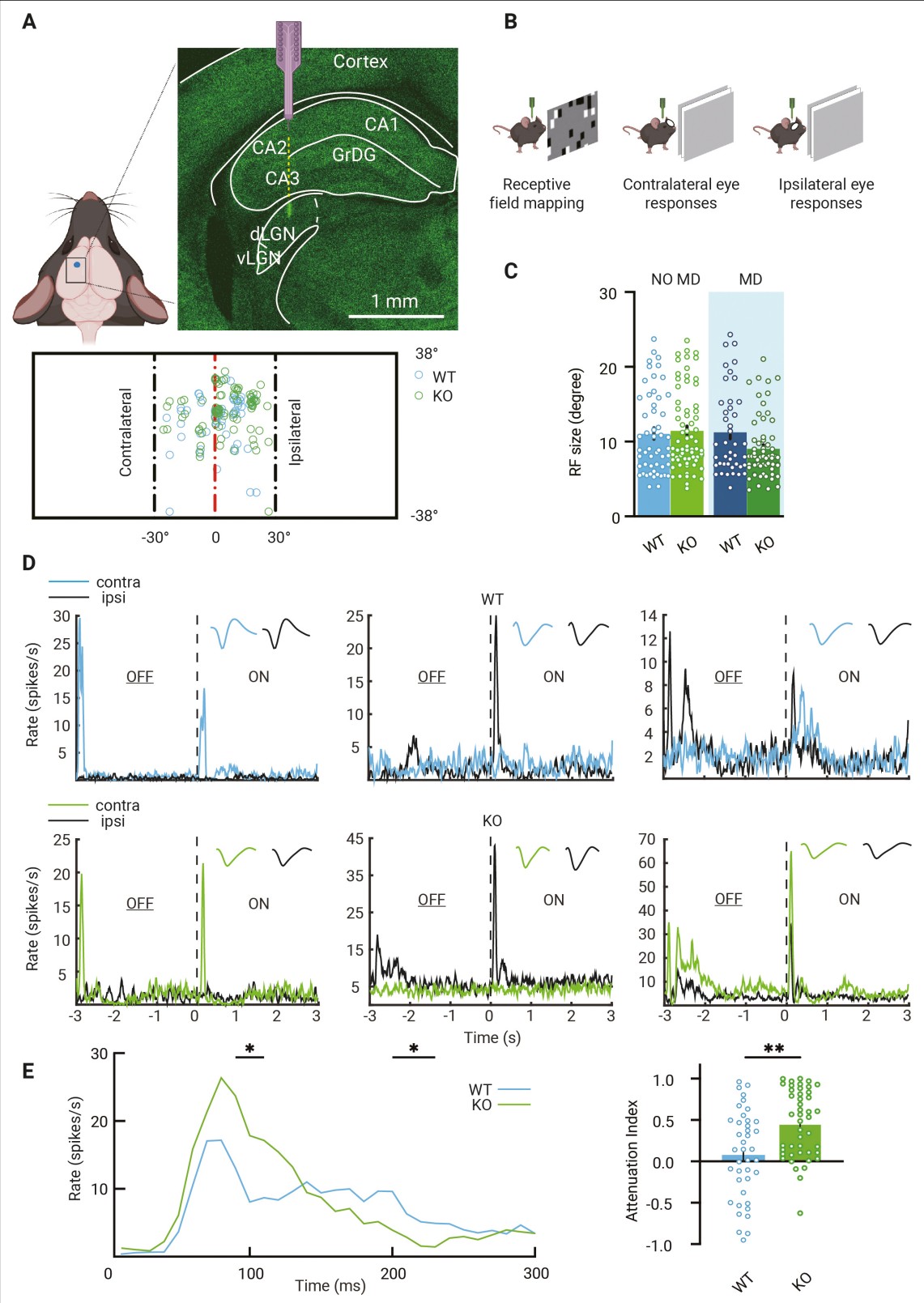

**Figure 1.** Visual responses of dorsolateral geniculate nucleus (dLGN) neurons in mice lacking thalamic *Gabra1*. (**A**) Recording electrodes were placed in the ipsilateral projection zone of dLGN (see green fluorescent trace of actual electrode penetration in dLGN). All receptive field (RF) centers of multiunits recorded in wild-type (WT, blue) and *Gabra1* cKO (KO, green) mice (n = 61 units from 13 non-deprived or monocularly deprived [MD] mice and n = 80 units from 18 NO MD or MD mice). Nose position is at 0° horizontally and vertically. The black dashed lines indicate –30° and +30° horizontal

*Figure 1 continued on next page*

*Figure 1 continued*

angles. (**B**) Experimental setup to measure RF and single-eye responses. (**C**) RF sizes of multiunits in NO MD and MD (shaded area) *Gabra1* cKO and WT mice do not differ (two-way ANOVA, interaction of genotype with MD: p=0.07; Tukey's post hoc test; WT NO MD vs. *Gabra1* cKO NO MD: p=0.19; WT MD vs. *Gabra1* cKO MD: p=0.11). (**D**) Examples of dLGN neuron responses to full-screen OFF-ON flash stimuli in WT and *Gabra1* cKO mice. Colored and black lines indicate responses of contra- and ipsilateral eyes, respectively. Waveforms of each unit responding to the contra- or ipsilateral eye are shown in the upper-right corner. WT: left panel is a monocular unit, right panels are binocular units. *Gabra1* cKO: left two panels are monocular units, right panel is a binocular unit (ZETA test). (**E**) Left, average responses of contralateral eye in WT and *Gabra1* cKO mice. *Gabra1* cKO mice show higher peak (90–110 ms) and lower prolonged responses (200–230 ms ). (Repeated-measure two-way ANOVA, interaction of genotype with time, p=0.0001; post hoc, Fisher's LSD test.) Right, attenuation index of visual responses in WT and *Gabra1* cKO mice. * p<0.05, ** p<0.01 (C, E) Error bars indicate standard error of the mean.

The online version of this article includes the following figure supplement(s) for figure 1:

**Figure supplement 1.** No differences in GAD67 or VACht puncta in dLGN of wild-type mice and mice lacking thalamic Gabra1.

observed during the phase of prolonged firing (*Figure 1E*). Thus, while average response strength in *Gabra1* cKO mice was similar to that in WT mice, the attenuation index was increased. These results show that like the situation during the critical period (*Sommeijer et al., 2017*) visual responses in dLGN neurons in adult *Gabra1* cKO and WT mice mostly differ in their temporal profile, while average response strengths and receptive fields sizes are hardly changed.

## OD plasticity in dLGN is reduced in adult mice lacking thalamic synaptic inhibition

We then continued experiments to assess OD plasticity in the dLGN of WT and *Gabra1* cKO mice. We monocularly deprived adult WT and *Gabra1* cKO mice for 7 d, long enough to induce an OD shift in adult V1 (*Frenkel and Bear, 2004*; *Heimel et al., 2007*; *Hofer et al., 2006*; *Lehmann and Löwel, 2008*; *Sato and Stryker, 2008*), by suturing one eye closed. We recorded responses to the ipsi- and contralateral eye in dLGN neurons (*Figure 2A*) and calculated the OD index (ODI) of all units recorded in monocularly deprived and non-deprived *Gabra1* cKO and WT mice and averaged them to obtain an OD score (*Figure 2B*). We found that after 1 wk of MD a significant OD shift occurred in dLGN of adult WT mice. This was predominantly caused by a significant increase in the responses to the non-deprived ipsilateral eye (*Figure 2C*). In *Gabra1* cKO mice, no OD shift could be induced in dLGN (*Figure 2B*) and no significant changes were observed in the responses to the ipsi- or contralateral eye (*Figure 2D*). Together, these results show that also in adulthood OD plasticity in dLGN depends on thalamic synaptic inhibition.

## OD plasticity in adult V1 is reduced in mice lacking thalamic synaptic inhibition

During the critical period, OD plasticity in V1 is partially deficient in *Gabra1* cKO mice. Brief MD induces a normal OD shift, but longer MD does not cause the OD shift to strengthen further (*Sommeijer et al., 2017*). This suggests that the critical period opens normally in V1 of *Gabra1* cKO mice, but that the second, homeostatic phase of the OD shift depends on thalamic inhibition and plasticity. Residual OD plasticity in adult V1 has various similarities with the second phase of OD plasticity during the critical period and also requires long-term MD. We therefore investigated whether OD plasticity in V1 was deficient in adult *Gabra1* cKO mice.

To address this question, we monocularly deprived WT and *Gabra1* cKO mice for 7 d, after which we recorded visual responses from single units in V1 of these mice and of normally sighted siblings (*Figure 3A*). Again, we only included channels with receptive fields within the central 30° of the visual field to ascertain we recorded from binocular V1. Like in dLGN, receptive field sizes did not differ between *Gabra1* cKO and WT mice and were not affected by a week of MD (*Figure 3B*).

Visual responses to the contra- or ipsilateral eye were recorded in the same way as for dLGN using the same visual stimuli (examples shown in *Figure 3C*). Like in dLGN, visual responses in V1 were more attenuated in *Gabra1* cKO mice than in WT mice (*Figure 3D*). Next, we calculated the ODI from all single units in the four groups (*Figure 3E*). A clear OD shift was induced in V1 of monocularly deprived WT mice. As expected, the OD shift involved an increase in open, ipsilateral eye responses (*Figure 3F*). We also found a significant decrease of deprived, contralateral eye responses (*Figure 3F*). While several studies have provided evidence that a loss of contralateral eye

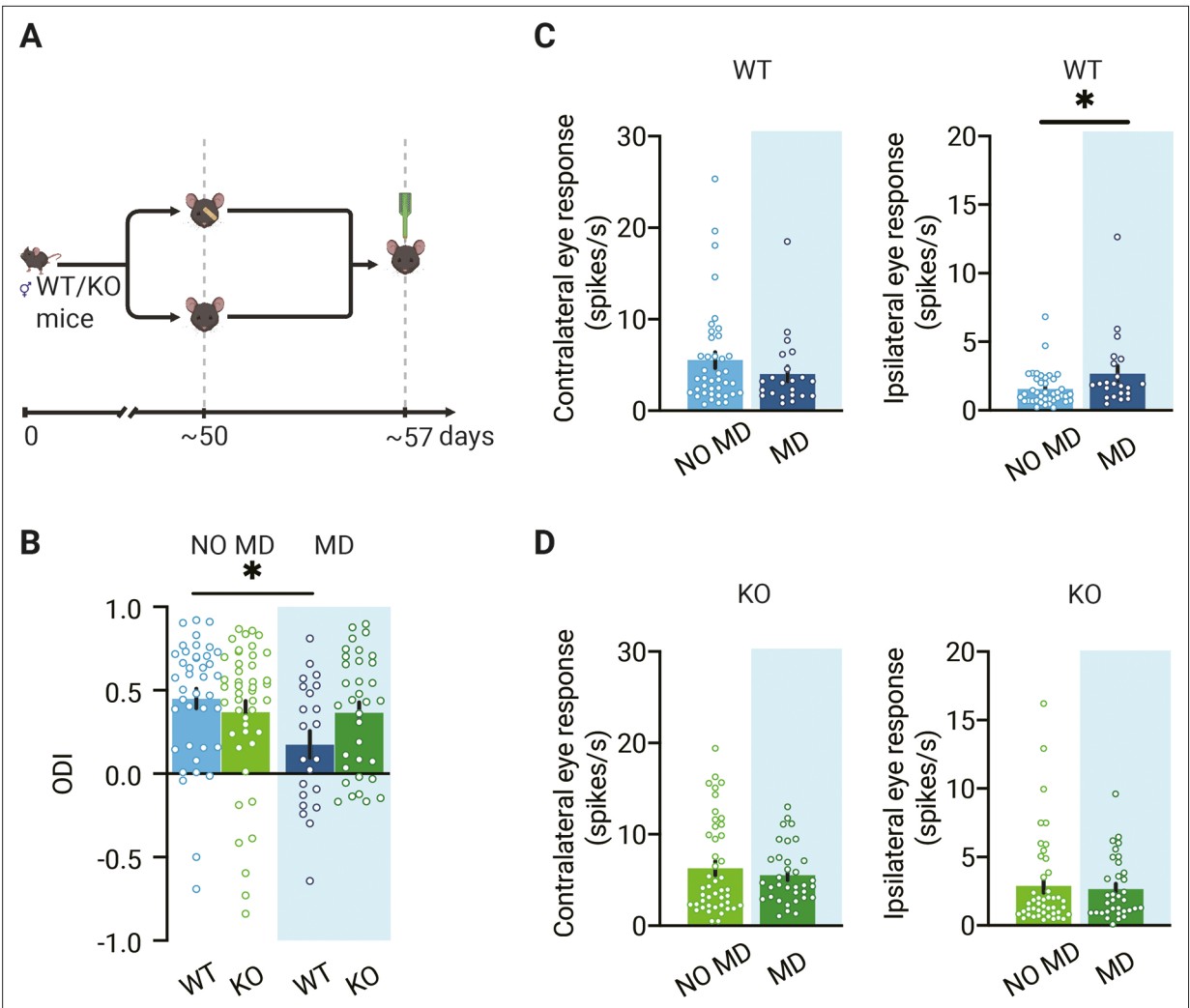

**Figure 2.** Loss of ocular dominance (OD) plasticity in dorsolateral geniculate nucleus (dLGN) of mice lacking thalamic synaptic inhibition. (**A**) Illustration of the experiment design. In experiments, four groups of animals were used: deprived (MD) or non-deprived (NO MD) wild-type (WT, blue) and *Gabra1* cKO (KO green) mice. Mice in the MD group had the eyelids of the eye contralateral to the recording side sutured for 7 d. (**B**) Seven days of MD reduces the ODI in WT mice but not in *Gabra1* cKO animals (interaction of genotype with MD: two-way ANOVA, p=0.046, Tukey's post hoc test; WT NO MD vs. WT MD, p=0.040; WT NO MD, n = 40 units, seven mice; WT MD, n = 22 units, six mice; *Gabra1* cKO NO MD, n = 45 units, nine mice; *Gabra1* cKO MD, n = 34 units, nine mice). (**C**) In WT mice, responses to the ipsilateral eye are significantly increased after 7-day MD. Responses to the contralateral eye are unchanged (Mann–Whitney; contralateral, NO MD vs. MD, p=0.29; ipsilateral, NO MD vs. MD, p=0.032). (**D**) In *Gabra1* cKO mice, MD causes no significant changes in responses to either the contralateral or the ipsilateral eye (Mann–Whitney; contralateral, NO MD vs. MD, p=0.73; ipsilateral, NO MD vs. MD, p=0.59). (B-D) Error bars represent standard error of the mean. * p<0.05.

responses contributes less to adult OD plasticity than during the critical period (*Frenkel and Bear, 2004*; *Kalogeraki et al., 2017*; *Sato and Stryker, 2008*), others have shown that it still occurs in adulthood (*Rose et al., 2016*). Possibly, the OD shift and the loss of deprived eye responses are more pronounced in our recordings due to them being limited to the center of the visual field or the use of flash stimuli instead of moving gratings. In adult *Gabra1* cKO mice, the OD shift after 7 d of MD was negligible and significantly smaller than in WT mice (*Figure 3G*). The contra- and ipsilateral eye responses in V1 of non-deprived *Gabra1* cKO mice were of comparable strength as those observed in WT mice, despite the lack of synaptic inhibition in the thalamus. After 7 d of MD, no significant strengthening of ipsilateral eye responses or weakening of deprived eye responses occurred (*Figure 3F*). We conclude that in adult mice absence of synaptic inhibition in the thalamus reduces OD plasticity in V1.

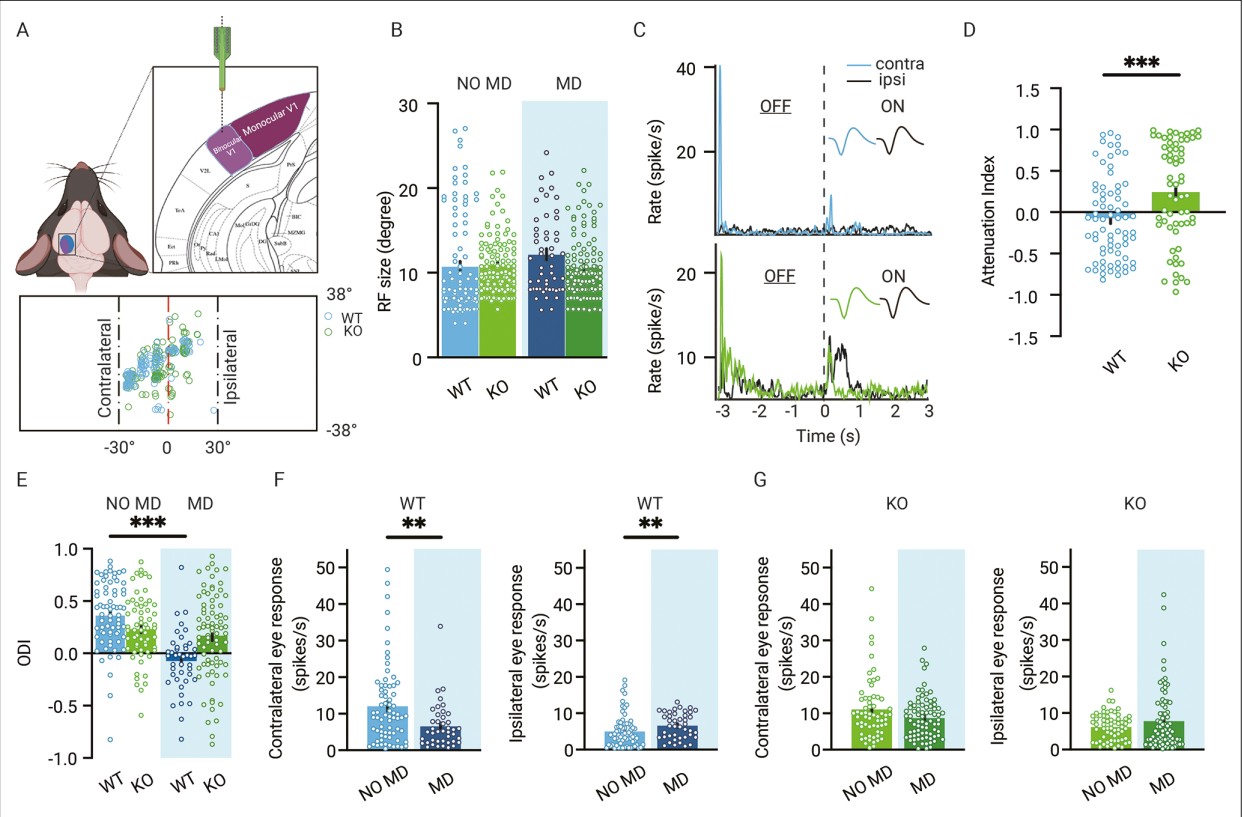

**Figure 3.** Reduced ocular dominance (OD) plasticity in adult V1 lacking thalamic OD plasticity. (**A**) Recording electrodes are located in binocular V1. All receptive field (RF) centers of multiunits recorded in wild-type (WT, blue) and *Gabra1* cKO (KO, green) mice (n = 112 units from 13 NO MD or MD mice and n = 138 units from 18 NO MD or MD mice). Nose position is at 0° horizontally and vertically. The black dashed lines indicate –30° and +30° horizontal angles. (**B**) RF sizes of units in WT and *Gabra1* cKO mice do not differ (interaction of genotype with MD: two-way ANOVA, p=0.07). (**C**) Two examples of single-unit responses in V1 of a WT and *Gabra1* cKO mouse to the contra- and ipsilateral eyes to ON and OFF visual stimuli. Each stimulus lasted 3 s. Colored and black lines indicate contra- and ipsilateral eye responses, respectively. (**D**) Attenuation index of contralateral eye responses in V1 of WT and *Gabra1* cKO mice. (**E**) Seven days of MD reduces the OD index (ODI) in WT but not *Gabra1* cKO mice (interaction of genotype with MD: two-way ANOVA, p<0.001, Tukey's post hoc test; WT NO MD vs. WT MD, p<0.001; WT NO MD, n = 71 units, seven mice; WT MD, n = 42, six mice; *Gabra1* cKO NO MD, n = 63 units, nine mice; *Gabra1* cKO MD, n = 78 units, nine mice). (**F**) In WT mice, responses to the contralateral eye are significantly reduced after 7-day MD, while those to the ipsilateral eye are significant increased (Mann–Whitney; contralateral, NO MD vs. MD, p=0.0043; ipsilateral, NO MD vs. MD, p=0.0062). (**G**) In *Gabra1* cKO mice, MD causes no significant changes in responses to either the contralateral or the ipsilateral eye (Mann–Whitney; contralateral, NO MD vs. MD, p=0.17; ipsilateral, NO MD vs. MD, p=0.66). (B, D, E-G) Error bars represent standard error of the mean. ** p<0.01, *** p<0.001.

## Effect of feedback from V1 to dLGN responses in the presence or absence of thalamic synaptic inhibition

These results so far show that OD plasticity in dLGN affects the OD shift in V1. OD plasticity in V1 may also influence the OD shift in dLGN. Apart from the retinal input dLGN relay cells receive, they are also strongly innervated by excitatory feedback connections from layer 6 cells in V1. Additionally, dLGN neurons receive bisynaptic inhibitory feedback from V1 via the thalamic reticular nucleus (TRN) and local interneurons (*Figure 4A*). Depending on whether excitatory or inhibitory feedback dominates, responses of dLGN relay cells to the ipsi- or contralateral eye in dLGN may be strengthened or inhibited by V1 feedback (*Denman and Contreras, 2015*; *Howarth et al., 2014*; *Jaepel et al., 2017*; *Kirchgessner et al., 2020*; *Olsen et al., 2012*). These feedback inputs from V1 can thus influence the OD of relay cells in dLGN.

To investigate how dLGN responses were influenced by V1 feedback and how synaptic thalamic inhibition affected this, we silenced V1 of WT and *Gabra1* cKO mice with the GABA-receptor agonist muscimol while recording from dLGN. Muscimol injections effectively silenced V1 (*Figure 4A*). On average, silencing V1 did not alter responses to the contra- or ipsilateral eye in individual units in

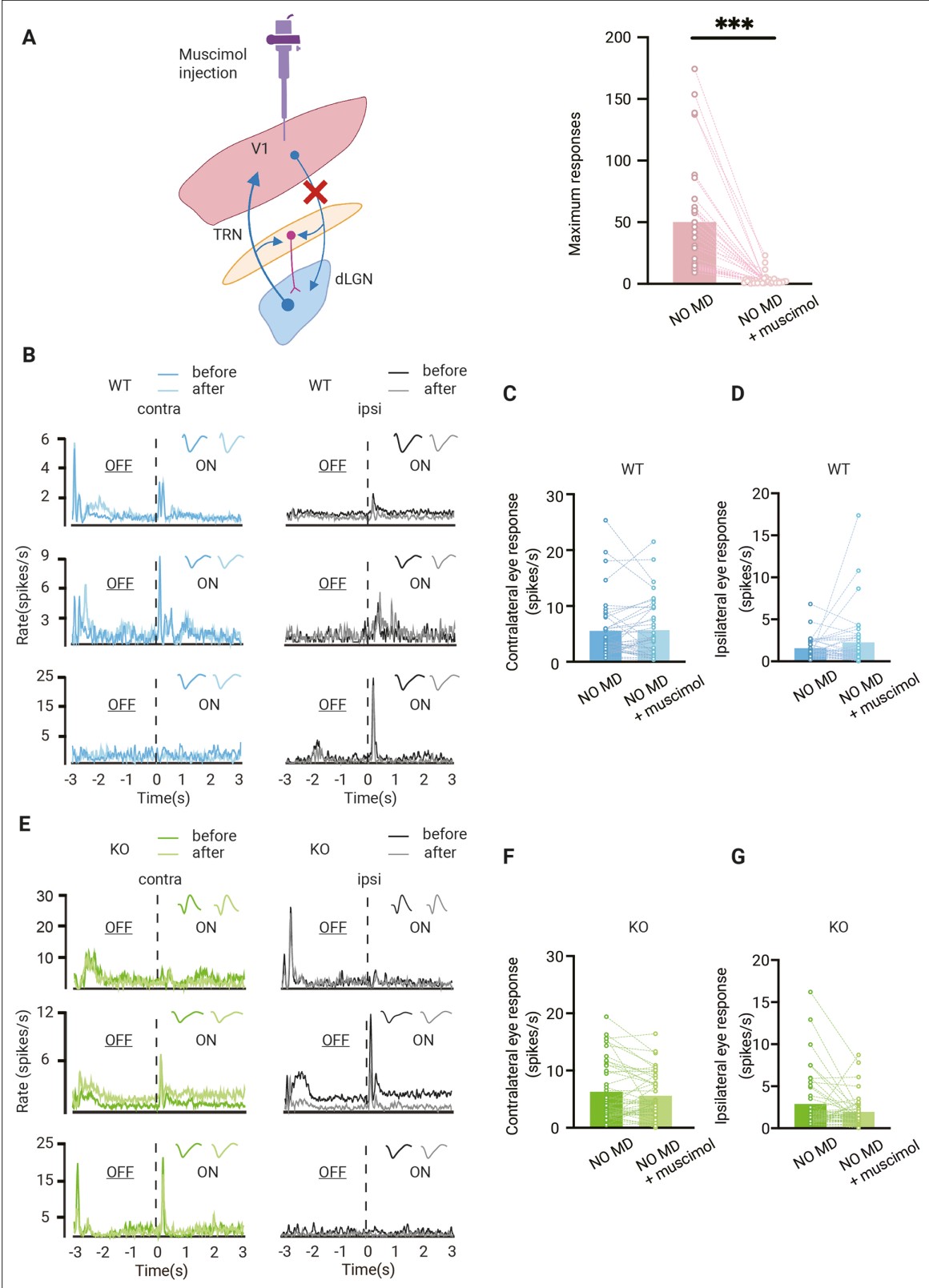

**Figure 4.** Effect of feedback from V1 to dorsolateral geniculate nucleus (dLGN) responses. (**A**) Left, illustration of corticothalamic-thalamocortical feedback network. dLGN is innervated by V1 and receives glutamatergic feedback. All these projections send excitatory collaterals to the thalamic reticular nucleus (TRN) which sends inhibitory inputs to dLGN. By muscimol injection in V1, corticothalamic projections are silenced. Right, V1 is effectively silenced by muscimol injection (Wilcoxon signed rank, p<0.001, n = 31 mice). (**B**) Examples of dLGN responses before and after muscimol

*Figure 4 continued*

injection in V1 of non-deprived wild-type (WT, blue) mice. Waveforms of each unit are shown in the upper-right corner. Left and right panels correspond to contralateral and ipsilateral eye responses, respectively. Dark and light lines represent responses before and after muscimol injection, respectively. (**C, D**) Silencing V1 feedback has no significant effect on contralateral (**C**) or ipsilateral (**D**) responses in WT mice (Wilcoxon signed rank; contralateral, WT NO MD vs. WT NO MD with muscimol, p=0.62; ipsilateral, WT NO MD vs. WT NO MD with muscimol, p=0.94, n = 40 units, seven mice). (**E**) Examples of dLGN responses before and after muscimol injection in V1 of non-deprived *Gabra1* cKO (KO, green) mice. Waveforms of each unit are shown in the upper-right corner. Left and right panels correspond to contralateral and ipsilateral eye responses, respectively. Dark and light lines represent responses before and after muscimol injection, respectively. (**F, G**) There is no significant effect of V1 silencing on contralateral (**F**) or ipsilateral (**G**) eye responses in *Gabra1* cKO mice, but a trend towards decreased ipsilateral eye responses is present (Wilcoxon signed rank; contralateral, *Gabra1* cKO NO MD vs. *Gabra1* cKO NO MD with muscimol, p=0.19; ipsilateral, *Gabra1* cKO NO MD vs. *Gabra1* cKO NO MD with muscimol, p=0.059, n = 45 units, nine mice).

dLGN of adult WT mice (*Figure 4B–D*). In *Gabra1* cKO mice, V1 silencing also did not significantly affect responses to the contralateral eye (*Figure 4E–G*). Responses to the ipsilateral eye showed a trend towards weakening after silencing V1, but this did not reach significance (p=0.094). Thus, also in the absence of synaptic inhibition in the thalamus, V1 feedback has relatively little influence on dLGN responses to the contra- or ipsilateral eye.

## Feedback from V1 does not affect the OD shift in adult dLGN

Finally, we investigated whether feedback from V1 influenced the OD shift in dLGN of adult WT and *Gabra1* cKO mice. In non-deprived mice, the ODI did not change after silencing V1 of WT mice (*Figure 5A*), as expected considering that contra- and ipsilateral eye response strengths were not affected by V1 feedback (*Figure 4C and D*). Similarly, the ODI in non-deprived *Gabra1* cKO mice did not change upon silencing V1 (*Figure 5B*), which was also expected based on the small changes in contra- and lateral eye responses that we observed (*Figure 4F and G*).

Despite the considerable OD shift we observed in V1 of adult WT mice, silencing V1 did not affect the OD measured in dLGN (*Figure 5C*). Also, average responses to the two eyes were not altered in the absence of cortical feedback (*Figure 5D and E*). This confirms that the OD shift in adult dLGN is not inherited from V1 (*Jaepel et al., 2017*) and supports the idea that dLGN plasticity involves the plasticity of retinogeniculate afferents. Interestingly, when we repeated this experiment in (C57Bl/6JRj) WT mice during the critical period, we found that silencing V1 also did not affect ipsi- or contralateral eye responses in non-deprived mice (*Figure 5—figure supplement 1*), but selectively reduced ipsilateral eye responses in monocularly deprived mice (*Figure 5G and H*). Consequently, silencing V1 significantly reduced the OD shift in these animals (*Figure 5I*). The change in ODI caused by V1 silencing in mice during the critical period was significantly larger than that in adult WT (p=0.034). Thus, during the critical period, corticothalamic connections strengthen the OD shift in dLGN, while they do not in adulthood.

In monocularly deprived adult *Gabra1* cKO mice (*Figure 5I–K*), silencing V1 did not affect responses to the contralateral eye, but significantly reduced those to the ipsilateral eye (*Figure 5J and K*), similarly to what we observed in WT mice during the critical period. However, despite this effect of V1 silencing, the average ODI in monocularly deprived *Gabra1* cKO mice was not significantly altered by it (*Figure 5I*). We conclude that feedback from V1 does not affect OD in the dLGN of adult mice, independently of whether they are monocularly deprived or lack synaptic inhibition in the thalamus. During the critical period, however, V1 silencing does reduce the OD shift observed in dLGN.

## Discussion

In summary, we show that OD plasticity in dLGN is reduced in adult mice lacking thalamic synaptic inhibition. In these mice, OD plasticity in V1 is absent, suggesting that it requires thalamic plasticity. We do not find evidence that feedback from V1 affects the thalamic OD shift in adult mice. This differs from the situation during the critical period, in which the OD shift in dLGN is partially inherited from V1.

In line with previous work (*Jaepel et al., 2017*), we find that in dLGN of adult mice, 7 d of MD causes a strong OD shift. The main substrate of this plasticity in adult dLGN appears to be the retinogeniculate synapse as silencing V1, the other main source of visual input to dLGN, does not affect the OD shift. Recent work has shown that while many relay cells in dLGN receive binocular inputs

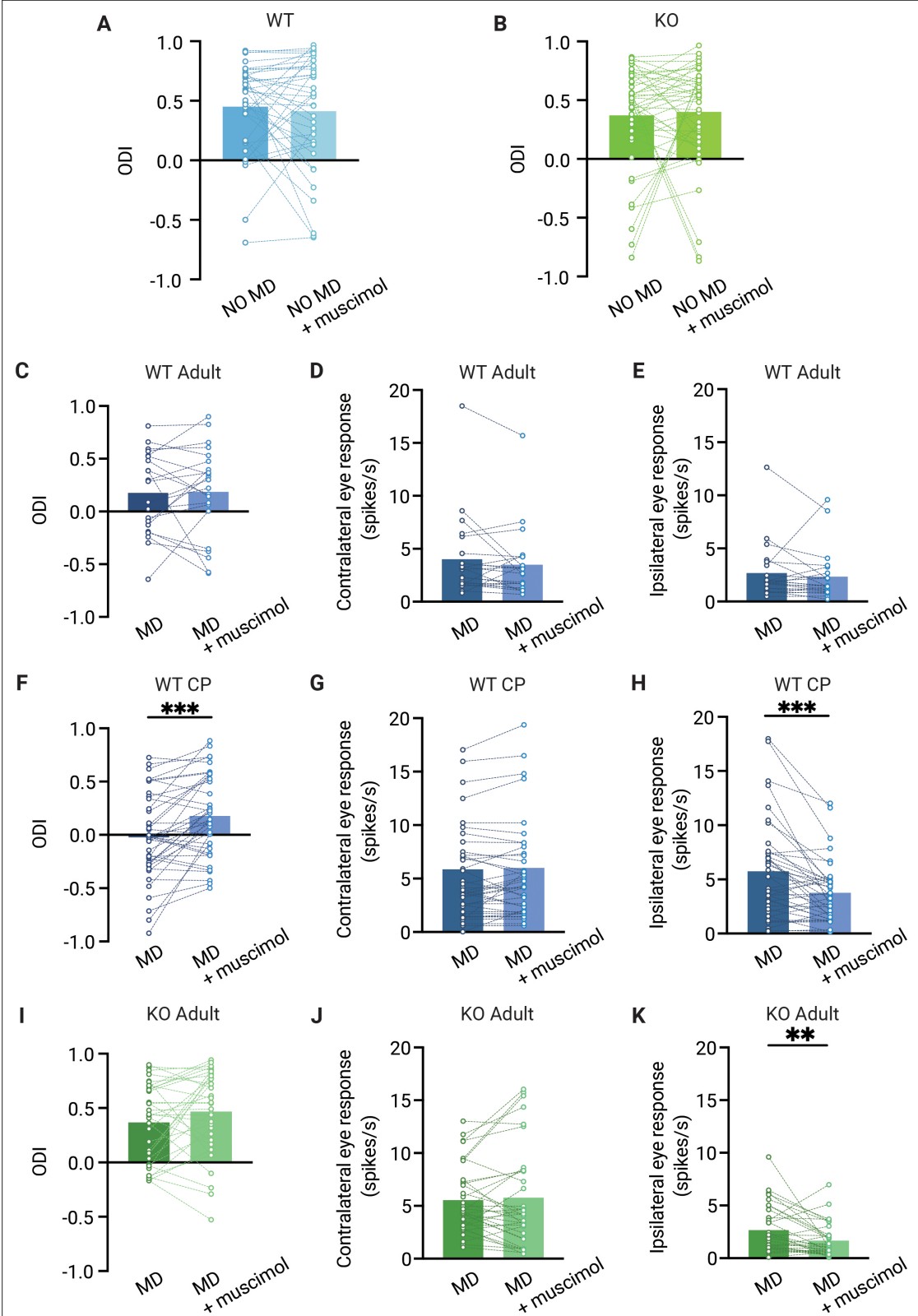

**Figure 5.** The ocular dominance (OD) shift in dorsolateral geniculate nucleus (dLGN) is independent from V1 feedback in adult mice but not in critical period mice. (**A, B**) Muscimol injection in V1 has no effect on the OD index (ODI) in dLGN of adult non-deprived wild-type (WT, blue) (**A**) and *Gabra1* cKO (KO, green) (**B**) mice (Wilcoxon signed rank; WT NO MD vs. WT NO MD with muscimol, p=0.86, n = 40 units, seven mice; *Gabra1* cKO NO MD vs. *Gabra1* cKO NO MD with muscimol, p=0.45, 45 units, nine mice). (**C–E**) Muscimol injection in V1 has no significant effect on the ODI (**C**), or contralateral

*Figure 5 continued on next page*

*Figure 5 continued*

(**D**) or ipsilateral (**E**) eye responses in dLGN of monocularly deprived WT mice (Wilcoxon signed rank; ODI, WT MD vs. WT MD with muscimol, p=0.89; contralateral, WT MD vs. WT MD with muscimol, p=0.21; ipsilateral, WT MD vs. WT MD with muscimol, p=0.10, n = 22 units, six mice). (**F–H**) During the critical period, V1 silencing has a significant influence on the ODI (**F**) and ipsilateral eye responses (**H**) in dLGN of monocularly deprived WT mice, but not on contralateral eye responses in these mice (**G**) (Wilcoxon signed rank; ODI, WT MD vs. WT MD with muscimol, p<0.001; contralateral, WT MD vs. WT MD with muscimol, p=0.46; ipsilateral, WT MD vs. WT MD with muscimol, p=0.003, n = 41 units, 10 mice). (**I–K**) Muscimol injection in V1 has no significant influence on the ODI and contralateral eye responses in *Gabra1* cKO MD mice, but significantly modulates dLGN ipsilateral eye responses (Wilcoxon signed rank; ODI, *Gabra1* cKO MD vs. *Gabra1* cKO MD with muscimol, p=0.13; contralateral, *Gabra1* cKO MD vs. *Gabra1* cKO MD with muscimol, p=0.97; ipsilateral, *Gabra1* cKO MD vs. *Gabra1* cKO MD with muscimol, p=0.004, n = 34 units, nine mice). ** p< 0.01, *** p<0.001.

The online version of this article includes the following figure supplement(s) for figure 5:

**Figure supplement 1.** Ocular dominance (OD) plasticity and the effect of cortical silencing on OD during the critical period.

(*Rompani et al., 2017*), synapses from the non-dominant eye are often silent and dominated by NMDA receptors (*Bauer et al., 2021*). Because silent synapses contribute strongly to OD plasticity in V1, it is interesting to speculate that OD plasticity in dLGN also involves the unsilencing and strengthening of these synapses (*Huang et al., 2015*; *Yusifov et al., 2021*). Indeed, the OD shift in adult mice involves many (monocular) contralateral eye-selective neurons to become binocular (*Jaepel et al., 2017*).

In apparent contradiction to our findings and the work by Jaepel et al., a recent study (*Li et al., 2023*) did not observe an OD shift in dLGN of adult mice upon 6 d of MD. A possible explanation for this apparent discrepancy is that Li et al. only included binocular dLGN neurons in the analysis. This approach will exclude the monocular, contralateral-eye selective neurons in non-deprived animals from the analysis, thus decreasing the measured OD shift.

We find that in adult *Gabra1* cKO mice OD plasticity in dLGN is absent. As we have previously shown that OD plasticity is also absent in dLGN of *Gabra1* cKO mice during the critical period, it is possible that maturation of dLGN is halted in a pre-critical period like state in adult *Gabra1* cKO mice, in analogy to the situation in GAD65-deficient mice in which reduced GABA release interferes with the onset of the critical period of cortical OD plasticity (*Fagiolini and Hensch, 2000*). However, we did not find evidence for halted thalamic development. Receptive fields of dLGN relay cells become smaller between eye opening and critical period onset (*Tschetter et al., 2018*), and we find that in WT and *Gabra1* cKO mice receptive field sizes are the same. Furthermore, there is a substantial increase in inhibitory and cholinergic boutons during this developmental stage (*Bickford et al., 2010*; *Sokhadze et al., 2018*; *Sommeijer et al., 2017*), but again, we find that adult WT and *Gabra1* cKO mice are not different in this respect. The primary difference between WT and *Gabra1* cKO mice in adult dLGN thus appears to be the lack of synaptic inhibition. We do not know how this lack of synaptic inhibition in adult dLGN interferes with OD plasticity, but a possible explanation may lie in altered temporal profile of the visual responses it causes, which could interfere with spike timing-dependent plasticity.

How does inactivation of synaptic inhibition and OD plasticity in dLGN interfere with the OD shift in V1? We believe that it is unlikely that OD plasticity in V1 is deficient due to halted development of V1 in a pre-critical period like state. We have previously shown that brief MD induces an OD shift in V1 of these mice (*Sommeijer et al., 2017*), suggesting that the critical period starts normally in *Gabra1* cKO mice. The previously observed plasticity deficit during the critical period was limited to the second stage of the OD shift in V1, which requires long-term MD. It thus appears that both during the critical period and in adulthood, the late phase of OD plasticity in V1 induced by long-term MD requires thalamic plasticity or inhibition. The most straightforward explanation for the cortical plasticity deficit is that MD-induced changes in dLGN relay cell responses to inputs from the two eyes contribute to the OD shift in V1. Additionally, the strengthening of responses to the non-deprived eye in dLGN neurons may provide their axons with a competitive advantage during OD plasticity in V1, further enhancing the OD shift in V1. Finally, it possible that OD plasticity in V1 is also affected by deficits in spike timing-dependent plasticity due to the more attenuated nature of dLGN responses that we observe in *Gabra1* cKO mice. Together, our results indicate that thalamic inhibition and plasticity play a crucial role in OD plasticity in adult V1, regardless of a possible developmental contribution to the plasticity deficit in dLGN.

The study by *Bauer et al., 2021* showed that binocularity in mouse dLGN may be lower than suggested by the current study and earlier work (*Howarth et al., 2014*; *Li et al., 2023*; *Sommeijer*

*et al., 2017*) that involved multi-electrode recordings in dLGN. Although this difference may be caused by technical limitations of single-unit recordings or calcium imaging, we think it is most likely explained by the fact that studies employing electrophysiological recording in dLGN targeted the frontal ipsilateral projection zone of dLGN, which is its most binocular region (*Bauer et al., 2021*). Recording in this region is essential when studying OD plasticity or binocularity in dLGN, but will strongly bias towards binocularly responding relay cells. When using two-photon imaging of dLGN boutons in V1 (*Bauer et al., 2021*; *Huh et al., 2020*; *Jaepel et al., 2017*), neurons from other parts of dLGN including the monocular shell- and caudal regions are also sampled.

It is unknown whether adult thalamic OD plasticity also occurs in species in which retinal inputs from the two eyes are organized in more strictly separated layers in dLGN, such as cats or primates. Studies in cats consistently found that upon MD or squint during the critical period the layers responding to the affected eye were thinner (*Hickey et al., 1977*; *Wiesel and Hubel, 1963b*) and that the neuronal responses in these layers were slower or weaker (*Eysel et al., 1979*; *Ikeda and Wright, 1976*; *Sestokas and Lehmkuhle, 1986*; *Wiesel and Hubel, 1963b*). Also in human amblyopes it was noted that dLGN responses to the amblyopic eye were weaker (*Hess et al., 2009*). So far, studies on dLGN plasticity by prolonged visual deprivation in adulthood are missing, though it was noted that in human subjects who suffered from glaucoma, the dLGN layers representing the affected eye were thinner (*Yücel et al., 2001*). Future research will need to establish whether plasticity in dLGN in humans contributes to amblyopia, and whether it can be enhanced to treat the disorder. That enhancement of thalamic plasticity is in principle possible is shown by experiments in mice, demonstrating that inactivation of the nogo-66 receptor in thalamus allows recovery of reduced acuity in adult mice that were monocularly deprived during development (*Stephany et al., 2018*).

Despite extensive monosynaptic excitatory feedback and bisynaptic inhibitory feedback (through TRN) from V1, we found that feedback does not affect the average strength of response to ipsi- or contralateral eye stimulation and thus neither the measured OD shift in dLGN of adult mice. The absence of V1 on the OD shift in adult mice was also reported previously (*Jaepel et al., 2017*). However in that study, silencing V1 with muscimol reduced the imaged calcium responses in boutons of dLGN axons projecting to V1, suggesting that dLGN neurons became less responsive to inputs from both eyes. A possible explanation for this apparent discrepancy may be that muscimol also has a direct effect on thalamocortical inputs (*Liu et al., 2007*; *Wang et al., 2019*; *Yamauchi et al., 2000*), which may reduce calcium responses in synaptic boutons of dLGN neurons without actually reducing their spiking activity at the soma.

Studies involving optogenetic stimulation of layer 6 neurons find that feedback from V1 suppresses dLGN responses (*Denman and Contreras, 2015*; *Kirchgessner et al., 2020*; *Olsen et al., 2012*), though this differs per cell and changes with stimulus strength and frequency (*Kirchgessner et al., 2020*). When V1 feedback is silenced, however, the average strength of dLGN responses is not reduced in most studies (*Denman and Contreras, 2015*; *Howarth et al., 2014*; *Kirchgessner et al., 2020*). This suggests that broad optogenetic stimulation of layer 6 predominantly recruits inhibitory feedback, while visual stimulation provides either more balanced or more limited excitatory and inhibitory feedback to dLGN. In line with these previous studies, we find that silencing V1 does not significantly alter dLGN responses to either eye in WT mice. If the lack of effect of V1 silencing is due to balanced excitatory and inhibitory feedback, one would expect that in *Gabra1* cKO mice lacking thalamic synaptic inhibition, silencing V1 would cause a reduction of dLGN responses. However, such an effect in *Gabra1* cKO mice was only observed in dLGN responses to the ipsilateral eye and reached significance only in monocularly deprived mice. This suggests that V1 feedback affects dLGN responses to the ipsilateral eye more strongly than those to the contralateral eye. This may be the result of V1 receiving input from the ipsilateral eye through dLGN but also through callosal V1 inputs (*Cerri et al., 2010*).

We found that during the critical period the influence of V1 feedback on the OD shift in dLGN was much stronger. It is possible that inhibitory innervation of dLGN is still weaker during the critical period than in adulthood, causing excitatory feedback from V1 to dominate like in *Gabra1* cKO mice. Additionally, a stronger OD shift occurs in V1 during the critical period, adding to the strength of the cortical feedback representing the ipsilateral eye. Interestingly, an excitotoxic lesion of V1 was found to alter OD in dLGN during development and affect OD plasticity in dLGN at various ages (*Li et al., 2023*). This suggests that continuous crosstalk between thalamus and cortex during development

guides plasticity, possibly optimizing thalamocortical and corticothalamic connections. The continued absence of corticothalamic feedback is likely to have a much larger impact on dLGN plasticity than the acute silencing we performed.

During the critical period, an experience-dependent phase of retinogeniculate refinement takes place, probably optimizing direction-selective inputs from the retina (*Hooks and Chen, 2020*; *Rompani et al., 2017*; *Thompson et al., 2017*). This experience-dependent refinement, like OD plasticity in dLGN, also depends on feedback from V1 (*Thompson et al., 2017*; *Thompson et al., 2016*). It is thus possible that refinement of binocular inputs and direction-selective inputs in dLGN are one and the same process.

We conclude that dLGN retains a high level of plasticity in adulthood and has considerable influence on cortical plasticity. This plasticity may not be restricted to binocular responses, but could also be relevant for other forms of perceptual learning (*Yu et al., 2016*). The findings stress that a thalamic involvement needs to be considered in amblyopia and learning disabilities. Additionally, the results may help understanding brain disorders that are thought to involve dysfunctional thalamo-cortical circuits, ranging from attention-deficit disorder (*Wells et al., 2016*) to schizophrenia (*Benoit et al., 2022*; *Pinault, 2011*; *Pratt et al., 2017*). Future experiments focusing on changes in thalamic responses and their interaction with the cortex may provide exciting new insights into how the brain learns.

## Methods

### Animals

All mice used to assess OD plasticity in adulthood were bred from homozygous conditional *Gabra1*-deficient mice (Gabra1$^{fl/fl}$) (*Vicini et al., 2001*) crossed with homozygous *Gabra1*-deficient, heterozygous *Olig3*$^{Cre/+}$ knockin mice (*Vue et al., 2009*) (*Gabra1*$^{fl/fl}$ *Olig3*$^{Cre/+}$). Before cross-breading the lines, *Gabra1*$^{fl/fl}$ mice had been backcrossed to C57Bl/6JRj mice (Janvier) for at least six generations. *Olig3*$^{Cre/-}$ mice were crossed to C57Bl/6JRj mice for at least two generations, but should be considered mixed background. All animals were tested for unintended germline recombination of the Gabra1$^{fl}$ locus, and such mice were excluded from breeding or experiments. In our experiments, we used four groups of animals: monocularly deprived or non-deprived *Gabra1*$^{fl/fl}$ *Olig3*$^{Cre/-}$ mice and monocularly deprived or non-deprived *Gabra1*$^{fl/fl}$ *Olig3*$^{Cre/+}$ siblings (P45–P90). The experimenter was blind to the genotype of the mice until the end of the experiment. Mice used for OD plasticity experiments during the critical period were C57Bl/6JRj mice. All mice were housed in a 12 hr/12 hr dark/light cycle. Both male and female mice were used. Mice housing conditions were according to Dutch law. All experiments were approved by the institutional animal care and use committee of the Royal Netherlands Academy of Arts and Sciences under Central Committee Animal experiments (CCD) licenses AVD 80100 2017 1045, AVD 80100 2022 15934 and AVD 80100 2022 15935.

### Immunohistochemistry

Age-matched mice were anesthetized with 0.1 ml/g body weight Nembutal (Janssen) and perfused with 4% paraformaldehyde (PFA) in PBS (~50 ml per mouse) and post-fixed for 2–6 hr. Post fixation time was consistent between compared groups. Sections from dLGN of 50 μm were made by using a vibratome (Leica VT1000S). Mouse-α-GAD67 (1:500, Chemicon, MAB5406) was used to label inhibitory boutons and guinea pig-α-VaChT (1:500, SySy 139105) to label cholinergic boutons. Primary antibodies were visualized using Fluor 594 goat-α-mouse (1:1000, Invitrogen, A11032) and Alexa Fluor 488 goat-α-guinea pig (1:1000, Invitrogen, A11073). Free-floating sections were briefly washed in PBS followed by 1–2 hr blocking in PBS containing 5% normal goat serum and 0.1% Triton X-100 at room temperature (RT). Primary antibody incubation was performed overnight at 4°C in fresh blocking solution. Next, the sections were washed three times for 10 min in PBS with 0.1% Tween-20 (PBST) followed by secondary antibody incubation in fresh blocking solution for 1.5–2 hr at RT. After washing three times for 10 min in TBST, the sections were mounted on glass slides using Mowiol (Calbiochem/MerckMillipore), glass covered, and stored at 4°C.

## Confocal microscopy and data analysis

Sections were imaged using a confocal microscopy (Leica SP5) with constant gain and laser power across compared samples. Care was taken that no signal clipping was present. For quantification of GAD67 and VAChT puncta, images were taken with a 40× objective (2048 × 2048 resolution). Background fluorescence was subtracted with ImageJ. VAChT and GAD67 puncta were quantified using the 'SynQuant' ImageJ plugin (*Wang et al., 2019*), creating ROIs corresponding to synaptic puncta. For each image, we calculated the average size of identified puncta, the number of puncta per unit of area, and the percentage of image area identified as part of punctum.

## Monocular deprivation

The eyelids of the eye contralateral to the recording side were sutured for MD. The surgery was performed under isoflurane anesthesia (5% induction, 1.5–2% maintenance in 0.7 l/min $O_2$). The eye was rinsed with saline. The eyelids were sutured together with 7.0 Ethilon thread. Eyes were checked for infection in the following days and reopened during recording. Only mice with healthy eye conditions were included.

## Electrophysiology recordings, visual stimulation, and V1 silencing

Mice were anesthetized by intraperitoneal injection of urethane (Sigma; 20% solution in saline, 1.2 g/kg body weight), supplemented by intraperitoneal injection of chlorprothixene (Sigma; 2.0 mg/ml in saline, 8 mg/kg body weight), followed by subcutaneous injection of atropine (Sigma; 1 mg/ml, 6 mg/kg body weight), and head-fixed by ear- and bite bars. The temperature was measured with a rectal probe and maintained at 36.5°C. The craniotomies for dLGN (2.0 mm lateral, 2.5 mm posterior from bregma) and V1 (2.95 mm lateral, 0.45 mm anterior from lambda) recording windows were made using a dental drill. During recordings of V1 or dLGN responses to input from one eye, the other eye was covered with a double layer of black cloth and black tape.

Using a linear silicon microelectrode (A1x16-5mm-25-177-A16, 16 channels spaced 50 µm apart, Neuronexus), extracellular recordings from V1 and dLGN were performed separately. Visual stimuli were projected by a gamma-corrected projector (PLUS U2-X1130 DLP) on a back-projection screen (Macada Innovision, covering a 60 × 42 cm area) positioned 17.5 cm in front of the mouse. The visual stimuli were programmed using the MATLAB (MathWorks) scripts package Psychophysics Toolbox 3 (*Brainard, 1997*). V1 was first recorded at a depth of approximately 800 µm from the cortical surface. Receptive field position was checked by showing white squares (5°) at random positions on a black background. If the receptive field was not within 30° from the center, we relocated the electrode and checked again. ODI was measured by presenting alternating white and gray full-screen stimuli to each eye in turn. Each stimulus lasted 3 s. Both white and gray screens were presented with 100 repetitions. When the V1 recording was finished, we relocated the electrode to dLGN at a depth of 2700–3000 µm from bregma. The receptive field and ODI measurement procedures were repeated in dLGN. We then silenced V1 by injecting muscimol (Sigma; 10 mM; ~150 µL per mouse), a selective agonist for $GABA_A$ receptors, in V1 and measured the ODI again in dLGN. After recording, we moved the electrode back to V1 to verify that muscimol had silenced V1.

If mice died before the full recordings were finished, the animal was excluded from analysis. Additionally, if analysis revealed that the response to visual stimulation with closed shutters was more than 1 standard deviation above zero, the mouse was excluded from OD analysis.

The extracellular signals were amplified and bandpass filtered at 500 Hz to 10 kHz and digitized at 24 kHz using a Tucker-Davis Technologies RX5 Pentusa base station. The spike detection was done by a voltage thresholder at 3× s.d. online per recording or offline using the open-source sorting package KiloSort (*Pachitariu et al., 2016*). Spikes were sorted and clustered by either principal component analysis-based custom-written MATLAB scripts or integrated template matching-based KiloSort scripts.

## Analysis of electrophysiological data

Data analysis was done using custom-made MATLAB scripts (https://github.com/heimel/inVivoTools, copy archived at *Heimel, 2023*). Data and code are shared on OSF (https://osf.io/dbhgw/). For each 3 s stimulus-related activity, we treated the last 500 ms of the previous trail as baseline. Therefore, we defined the visual responses as the difference between the first 500 ms of the stimulus and the mean

of the last 500 ms activities of the previous stimulus. The peak visual responses of stimuli were considered as the maximum firing rates in the first 300 ms of visual-related responses. The visual responses were calculated as average responses of 300 ms. ODI was calculated as $\left(R_{contra} - R_{ipsi}\right) / \left(R_{contra} + R_{ipsi}\right)$, where the $R_{contra}$ is the average firing rate of the unit when contralateral eye was open and ipsilateral eye was covered; $R_{ipsi}$ is the opposite. For receptive field mapping, we computed the spike-triggered average of the random sparse squares stimulus. The peak rate threshold was set to 5 Hz when the patch was within the receptive field. The actual position and size of visual field were computed and corrected for the actual distance between stimuli and animal. For the assessment of the ODI, only included recordings from channels with receptive fields corresponding to the central 30° of the visual field were included.

To assess the monocular and binocular categorical responsiveness, we performed ZETA test (https://github.com/JorritMontijn/ZETA; *Montijn et al., 2021*; *Montijn, 2023*) over all trials on all units that we recorded. ZETA (Zenith of Event-based Time-locked Anomalies)-test is a more powerful and sensitive tool than *t*-test or ANOVA to detect whether a cell is responsive to stimulation in a statistically robust way and avoids binning and parameter selection together. The significant on- and off responses were calculated separately. Therefore, for each unit, four responses were tested: contra on, contra off, ipsi on, and ipsi off. For each unit, if the contra on or off response passed the ZETA test (p<0.05), but neither the ipsi on nor the off response did, we categorized this unit as pure contra. The same approach, but then reversely, was used to determine pure ipsilateral eye-responsive units. If a contralateral eye on or off response and an ipsilateral eye on or off response passed the ZETA test, we considered this unit as binocular.

## Statistics

For testing the interaction between genotype with MD on OD plasticity and RF size in V1 and dLGN, we used a two-way ANOVA test with post hoc Tukey–Kramer tests. Differences in the temporal profile of the visual responses in WT vs. *Gabra1* cKO mice were determined using a repeated-measure two-way ANOVA with post hoc Fisher's LSD test. Quantitation of immunohistochemical analyses and difference in OD shift upon cortical silencing were performed using Student's *t*-test. All other tests were done with non-parametric tests. Statistical analyses of the response changes of V1 and dLGN units in WT and *Gabra1* cKO mice were done by non-parametric Mann–Whitney *U*-tests. For testing the significance of the effect of silencing V1 on adult and critical period dLGN responses, on the OD in adult deprived and non-deprived *Gabra1* cKO mice and WT mice, on the OD in critical period deprived WT mice, and the effects of muscimol injection on V1 responses, Wilcoxon signed-rank tests were used.

## Acknowledgements

The authors thank Emma Ruimschotel for technical assistance, Robin Haak and Huub Terra for help with spike sorting, Huub Terra for providing feedback on the manuscript, and the Mechatronics department and Animal facilities of the NIN for their services. *Figures 1A&B, 2A, 3A and 4A* were created with BioRender.com (2022). This project has received funding from the NeuroTime Erasmus+ program of the European Commission and the European Union's Horizon 2020 Research and Innovation Programme under grant agreements no.785907 (HBP SGA2) and 945539 (HBP SGA3).

## Additional information

### Funding

| Funder | Grant reference number | Author |
| --- | --- | --- |
| European Commission | NeuroTime Erasmus+ | Christiaan N Levelt |
| European Commission | Human Brain Project 785907 (HBP SGA2) and 945539 (HBP SGA3) | Christiaan N Levelt |

The funders had no role in study design, data collection and interpretation, or the decision to submit the work for publication.

## Author contributions
Yi Qin, Conceptualization, Data curation, Software, Formal analysis, Investigation, Methodology, Writing – original draft; Mehran Ahmadlou, Paul Neering, Investigation, Methodology; Samuel Suhai, Formal analysis, Investigation, Methodology; Leander de Kraker, Software, Formal analysis; J Alexander Heimel, Formal analysis, Validation, Methodology; Christiaan N Levelt, Conceptualization, Resources, Supervision, Writing – original draft, Project administration, Writing – review and editing

## Author ORCIDs
Yi Qin (ID) http://orcid.org/0000-0002-8844-3097
J Alexander Heimel (ID) http://orcid.org/0000-0002-5291-4184
Christiaan N Levelt (ID) https://orcid.org/0000-0002-1813-6243

## Ethics
Mice housing conditions were according to Dutch law. All experiments were approved by the institutional animal care and use committee of the Royal Netherlands Academy of Arts and Sciences under Central Committee Animal experiments (CCD) licenses AVD 80100 2017 1045, AVD 80100 2022 15934 and AVD 80100 2022 15935.

Reviewer #1 (Public Review): https://doi.org/10.7554/eLife.88124.3.sa1
Reviewer #2 (Public Review): https://doi.org/10.7554/eLife.88124.3.sa2
Author Response https://doi.org/10.7554/eLife.88124.3.sa3

# Additional files

## Supplementary files
• MDAR checklist

## Data availability
Data and script to reproduce the figures can be found at https://osf.io/dbhgw/. This also requires installation of the custom-made MATLAB scripts used for data analysis (https://github.com/heimel/inVivoTools; copy archived at *Heimel, 2023*).

The following dataset was generated:

| Author(s) | Year | Dataset title | Dataset URL | Database and Identifier |
|---|---|---|---|---|
| Qin Y, Heimel A, Levelt C | 2023 | Electrophysiology data for 'Thalamic regulation of ocular dominance plasticity in adult visual cortex' | https://osf.io/dbhgw/ | Open Science Framework, dbhgw |

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
