## [Editor Report · eLife assessment]

This study demonstrates that plasticity of ocular dominance of binocular neurons in the visual thalamus persists in adulthood. The evidence supporting the authors' conclusion is **convincing**, and the findings are an **important** contribution to a growing body of work identifying plasticity in the adult visual system. This work will interest those in the field of ocular dominance plasticity in the visual system as well as scientists investigating the function of synaptic plasticity in the brain.

---

## [Referee Report · Reviewer #1 (Public Review)]

Qin et al., demonstrate, convincingly, that plasticity of ocular dominance of binocular neurons in the visual thalamus persists in adulthood. The adult plasticity is similar to that described in critical period juveniles in that it is absent in transgenic mice with the deletion of the GABA a1 receptor in thalamus, which also blocks ocular dominance plasticity in primary visual cortex. However, the adult plasticity is not dependent on feedback from primary visual cortex, an important difference from juveniles. These findings are an important contribution to a growing body of work identifying plasticity in the adult visual system, and identifies the visual thalamus as a potential target for therapies to reverse adult amblyopia.

---

## [Referee Report · Reviewer #2 (Public Review)]

In this work, the authors found in the mouse line of GABA a1 subunit KO in thalamic neurons, which was previously reported lacking ocular dominance (OD) plasticity in juvenile V1 and dLGN (Sommeijer et al., 2017), the adult V1 and dLGN OD plasticity was also missing. Through muscimol inhibiting the V1 feedback, thalamic OD plasticity was unaffected in both WT and KO adult mice. However, during the critical period, the thalamic OD plasticity was dependent on V1 feedback in WT mice.

Strengths:

1. The experiments were well designed. The authors used both MD and No MD controls with both WT and KO mice. The authors used in vivo SU recording, which is broadly accepted as the major method for evaluating OD plasticity.

2. The data analysis was solid. The authors used proper statistical tests for non-parametric data set.

Weaknesses:

1. In my previous review I pointed out that an alternative interpretation of the results is that the lack of OD plasticity in adult V1 and dLGN was caused by an early blockade of the development of the inhibitory circuit in dLGN, which causes life-long deficits in the functional connection of dLGN. The best way to rule out this possibility is by using conditional KO mice that dLGN synaptic inhibition was only interfered in adulthood. In response to my concern, the authors replied with a long text of reasoning why the current results are solid enough and the proposed experiment was unnecessary. I agree with most of the explanation that the current conclusion is solid, but I still think that the cKO experiment will be a good supplement to the current study, and if we do see a similar result in the cKO mice, the conclusion that the adult perturbation of thalamic inhibitory circuit interfere with the OD plasticity will be more convincing. However, I do understand that repeating the experiments again in another mouse line will be difficult and time-consuming, so the authors could choose if they want to perform the experiment or not.

2. Now the discussion part is very long and complex. Rearranging the discussion with sub-sections will make it easy to read.

---

## [Author Response]

The following is the authors’ response to the original reviews.

**Reviewer #1 (Recommendations For The Authors):**
Please describe the criteria for binocularity of dLGN neurons, and what % of recorded neurons meet this criteria. Do all the example neurons in figure 1D meet the criteria for binocular neurons?

We now include criteria for binocularity of dLGN neurons in the methods section on page 24, and mention the percentage of binocular neurons that we detected. We also indicate which of the example neurons in figure 1D are monocular or binocular according to these criteria. We would like to stress that these percentages are not representative for the level of binocularity in dLGN as a whole, as our recordings were limited to the frontal ipsilateral projection zone of dLGN, which is its most binocular region, and only units with a receptive field within 30o from the center were included in the analysis. We mention this in the discussion on page 23.

Fig 1: Please perform statistical comparison of data presented in Figure 1c by genotype, as in other figures.

We conducted post-hoc Tukey's tests exclusively when a significant interaction between phenotype and genotype was detected in the two-way ANOVA (as seen in Figs. 2B and 3E). This decision was made because interpreting a significant post-hoc test becomes uncertain when there is no interaction, which is evident in Fig. 1C. In that case, the posthoc Tukey's test yielded a p-value of 0.044 for the difference in RF size between KO NOMD and KO MD, while all other comparisons were not significant (WT NO-MD vs WT MD: P=0.15, WT NO-MD vs KO NO-MD: p=0.99, WT MD vs KO MD: p=0.21). However, since there was no significant interaction between genotype and phenotype, we cannot conclude that there is an effect in KO mice that is absent in WT mice. In Fig. 3B, all posthoc Tukey's tests resulted in P-values greater than 0.05.

Fig 1e: There is no justification for splitting the data into two time epochs before and after 150 msec. A repeated measures anova of smaller time bins across the full time course would be more effective/appropriate here.

The reviewer is correct. We have now performed a repeated measures ANOVA.

Fig 2: GABA a1R KO results in a loss/absence of OD plasticity, not a reduction

We agree. We have changed the wording.

Fig 3: Please be specific about the location of V1 recordings. Was layer-specificity determined?

The location of V1 recordings is mentioned in the methods section under“Electrophysiology recordings, visual stimulation and V1 silencing”, page 23. We have assessed OD per depth, but found that we do not have sufficient units to draw any conclusions about differences in plasticity per layer.

Why is feedback from V1 more influential in dLGN OD plasticity in KO?

We believe this is because the reduced thalamic inhibition causes the excitation/inhibition ratio to shift in favor of excitation. We discuss this more extensively on page 19 of the discussion.

Fig 4: Inclusion of a GABA R antagonist protects thalamic axons from muscimol silencing (Liu BH, Wu GK, Arbuckle R, Tao HW, Zhang LI. Defining cortical frequency tuning with recurrent excitatory circuitry. Nat. Neurosci. 2007;10:1594-600.)

We now mention the possible direct influence of muscimol on thalamic axons in the discussion on page 19 and cite the suggested article.

The observation that feedback from primary visual cortex does not contribute to adult visual thalamus plasticity is interesting and important. The authors should expand on their discussion of this observation to include changes in cortical circuitry that may help to explain this observation.

We have expanded this part of the discussion on page 20.

The authors should describe the pathway by which inhibition enables plasticity in dLGN.

We discuss this more extensively on page 17 in the updated manuscript.

**Reviewer #2:**
1. The current work was basically a follow-up of a previous study in juvenile mice, and the results were also very similar to the juvenile results (Sommeijer et al., 2017). One possible interpretation of the results is that the lack of OD plasticity in adult V1 and dLGN was caused by an early blockade of the development of the inhibitory circuit in dLGN, which retains the dLGN in an immature stage till adulthood. The authors indeed claimed in the discussion that the 2-day OD shift is intact in juvenile dLGN and V1 in KO mice, and provided evidence in supplementary figure that GABAergic and cholinergic synapse amount are similar between WT and KO mice. However, the 7-day OD shift is indeed defected in juvenile V1 and dLGN in KO mice (Sommeijer et al., 2017), and it is possible that this early functional deficit didn't induce a structural remodeling in adulthood. To better support the author's claim that the lack of adult V1 OD plasticity is specifically due to reduced dLGN synaptic inhibition, the author should generate conditional KO mice that dLGN synaptic inhibition was only interfered in adulthood.

In order to address this criticism it is important to discuss the plasticity deficits in dLGN and V1 separately.

Concerning V1 plasticity: We have previously shown that brief MD induces an OD shift inV1 of mice lacking thalamic synaptic inhibition in dLGN. OD plasticity induced by brief MD is a hallmark of critical period plasticity in V1, and it thus seems highly unlikely that critical period onset in V1 is defective or that development of V1 is halted in an immature state that does not support OD plasticity in thalamus-specific GABRA1 deficient mice.

The observed plasticity deficit during the critical period was limited to the second stage of the OD shift in V1, which requires long-term monocular deprivation. The straightforward explanation for this result and our current findings is that both during the critical period and in adulthood, the second stage of OD plasticity in V1 induced by long-term monocular deprivation requires thalamic plasticity or inhibition. The proposed alternative, that lack of thalamic synaptic inhibition during development results in a possible lack of structural change in V1 that would cause a lifelong deficiency selectively affecting OD plasticity induced by long-term monocular deprivation, requires many more assumptions.

Concerning dLGN plasticity: The simplest explanation for the observed lack of OD plasticity in dLGN is that it is a direct consequence of the absence of synaptic inhibition in the KO mice. However, an alternative explanation could indeed be that dLGN is kept in an immature (pre-critical period-like) state due to the developmental absence of synaptic inhibition. This situation would be analogous to that in V1 of GAD65 deficient mice (which have reduced GABA release), in which OD plasticity cannot be induced by brief monocular deprivation during the critical period or in adulthood (Fagiolini and Hensch, 2000). Because this deficit can be reversed by treating the mice with benzodiazepines (allosteric modulators of GABA receptors) at any age, it is thought that development of V1 in GAD65 mice is halted in a pre-critical period-like state until inhibition is strengthened. We cannot exclude that something similar occurs in dLGN of mice lacking thalamic synaptic inhibition, although we did not observe any changes in hallmarks of dLGN maturity, such as reduced receptive field size, and increased cholinergic and inhibitory bouton densities.

However, if the analogy with the developmental deficit in V1 of GAD65 deficient mice is valid, the reduced plasticity is still likely to be a direct consequence of reduced inhibition. In GAD65 deficient mice, long term monocular deprivation during the critical period causes a full OD shift, showing that no additional deficits (besides reduced inhibition) limit OD plasticity in V1 of these mice (Gagiolini and Hensch 2000). And, as already mentioned, increasing inhibition rescues OD plasticity in GAD65 KO mice. Thus, the immature state of V1 in these mice is probably nothing more than a situation in which inhibition tone is too low to support efficient OD plasticity. In dLGN, knocking out GABRA1 at a later age could therefore also create a situation in which inhibition is too low to support thalamic OD plasticity, which is not different from the situation in which the gene is inactivated at birth. Only if lack of synaptic inhibition in thalamus affects another, unknown developmental process that is of importance later in life to support OD plasticity in dLGN, the proposed experiment would result in a different outcome. We are not convinced that this scenario is likely enough to justify repeating most of this study, but now using mice in which GABRA1 is inactivated in dLGN through bilateral AAV-cre injections.

Independently of the exact cause of the plasticity deficit in dLGN, our results make clear that a cortical plasticity deficit in adulthood can have a thalamic origin, which we believe is an important insight that is highly relevant.

We have included part of these arguments in the discussion on page 17.

1. The authors found that in juveniles, dLGN OD shift is dependent on V1 feedback, but not in adults. However, a recent work showed that the effects of V1 silencing on dLGN OD plasticity could differ with various starting points and duration of the V1 silencing and MD (Li et al., 2023). Could the authors provide more details of the MD and V1 silencing for an in-depth discussion?

We discuss some of the findings of the Li et al paper on pages 16 and 20 of the manuscript now.